# Ligand Tuning of Localized Surface Plasmon Resonances in Antimony-Doped Tin Oxide Nanocrystals

**DOI:** 10.3390/nano12193469

**Published:** 2022-10-04

**Authors:** Olexiy Balitskii, Oleksandr Mashkov, Anastasiia Barabash, Viktor Rehm, Hany A. Afify, Ning Li, Maria S. Hammer, Christoph J. Brabec, Andreas Eigen, Marcus Halik, Olesya Yarema, Maksym Yarema, Vanessa Wood, David Stifter, Wolfgang Heiss

**Affiliations:** 1Institute-Materials for Electronics and Energy Technology (i-MEET), Department of Materials Science and Engineering, Friedrich-Alexander-Universität Erlangen-Nürnberg, Energy Campus Nürnberg, Fürtherstraße 250, 90429 Nürnberg, Germany; 2Department of Electronics, Lviv Ivan Franko National University, Dragomanov Str., 50, 79005 Lviv, Ukraine; 3Adolphe Merkle Institute, Fribourg University, 1700 Fribourg, Switzerland; 4Institute-Materials for Electronics and Energy Technology (i-MEET), Department of Materials Science and Engineering, Friedrich-Alexander-Universität Erlangen-Nürnberg, Martensstraße 7, 91058 Erlangen, Germany; 5Department of Laser Sciences and Interactions, National Institute of Laser Enhanced Sciences (NILES), Cairo University, Giza 12613, Egypt; 6Helmholtz-Institut Erlangen-Nürnberg, Immerwahrstraße 2, 91058 Erlangen, Germany; 7Organic Materials & Devices, Department of Material Science, Interdisciplinary Center for Nanostructured Films (IZNF), Friedrich-Alexander University Erlangen-Nürnberg (FAU), Cauerstrasse 3, 91058 Erlangen, Germany; 8Institute for Electronics, Department of Information Technology and Electrical Engineering, ETH Zürich, 8092 Zürich, Switzerland; 9Center for Surface and Nanoanalytics (ZONA), Johannes Kepler University Linz, 4040 Linz, Austria

**Keywords:** nanocrystals, plasmonics, colloids, metal-oxides

## Abstract

Aliovalent-doped metal oxide nanocrystals exhibiting localized surface plasmons (LSPRs) are applied in systems that require reflection/scattering/absorption in infrared and optical transparency in visible. Indium tin oxide (ITO) is currently leading the field, but indium resources are known to be very restricted. Antimony-doped tin oxide (ATO) is a cheap candidate to substitute the ITO, but it exhibits less advantageous electronic properties and limited control of the LSPRs. To date, LSPR tuning in ATO NCs has been achieved electrochemically and by aliovalent doping, with a significant decrease in doping efficiency with an increasing doping level. Here, we synthesize plasmonic ATO nanocrystals (NCs) via a solvothermal route and demonstrate ligand exchange to tune the LSPR energies. Attachment of ligands acting as Lewis acids and bases results in LSPR peak shifts with a doping efficiency overcoming those by aliovalent doping. Thus, this strategy is of potential interest for plasmon implementations, which are of potential interest for infrared upconversion, smart glazing, heat absorbers, or thermal barriers.

## 1. Introduction

For many decades, localized surface plasmon resonances (LSPRs) were predominantly attributed to metal nanoparticles (NPs), wherein the free electron concentration is high enough to resonantly interact with electromagnetic radiation in visible and UV [1]. Nanocrystals (NCs) fabricated from more recently invented degenerate semiconductors widen the plasmon oscillations’ frequency range to extend into the near- and mid-infrared region [2,3], meeting demands in biology, medicine, electronics, and renewable energy. Among semiconductor LSPR materials, transparent conducting oxides (TCOs) are used in energy applications, i.e., as hydrogen evolution catalysts, solar cell sensitizers, in batteries, and supercapacitors, [4,5]. Those TCO NCs can be classified into two categories, depending on the strategies of how to provide high carrier concentrations, namely by self- and by aliovalent doping. The former exploits oxygen vacancies for doping. These materials are characterized by a lack of NCs’ stability in ambient conditions and by limited plasmon tuning ranges. The latter requires more complicated chemistry, but overcomes the self-doped NCs in their prolonged resistance against oxidation and more precise plasmon control. To date, the most well-known aliovalent-doped indium tin oxide (ITO) plasmonic NCs are widely used in commercial applications where low electrical resistivity and high absorbance of ‘waste’ IR energy are required, e.g., UV/Vis transparent glazing thermal heaters and deicers [6] and smart windows [7]. ITO plasmonic NCs are vital in solar energy harvesting, effectively utilizing two portions of solar spectra energies beyond visible (c.a. 1400–1800 and 2000–2500 nm). In hybrid devices, the coincidence of those bands with NCs’ LSPR contributes to the overall device efficiency by sub-bandgap absorbance in photovoltaic thermal module concentrators [8]. Extrinsically doped LSPR TCO materials [9] are more advantageous than self-doped plasmonic semiconductors [10] owing to their thermal stability up to boiling points of specific thermal fluids (c.a. 250 °C). The strong electric field enhancement in the proximity of LSPR NCs stimulates two-photon absorption processes in coupled dye molecules [11]. The two-photon absorption would be facilitated by fine tuning strategies for the near-infrared LSPRs—as the plasmonic absorbance or scattering peak has to match the double wavelength of the dye transition. The major drawback of ITO materials is the scarcity of indium, generally comparable to that of noble metals, but with much fewer recycling and recovery techniques available to date [12]. Recently invented fluoride and indium co-doped cadmium oxide [13] LSPR NCs are less indium consuming, but will presumably fail in meeting the RoHS requirements because even the current exceptions for cadmium (e.g., solar PV modules) are questionable to be renewed. Owing to the cheapness, facile processes of syntheses, and plasmonic properties closely matching the ITO materials, antimony-doped tin oxide (ATO) is a perfect substitute for the applications described above [14,15,16]. Despite belonging to an ultra-wide bandgap semiconductor class, tin dioxide (TO) possesses good thermal conductivity and carriers’ mobility; the aliovalent and self-doping of TO offers the opportunity for efficient carrier concentration control [17]. The Haase group introduced plasmonic ATO NCs [18] in 1999. The three-stage procedure of the ATO NCs’ preparation includes room-temperature synthesis of low-degree crystalline templates, their subsequent purification, and aqua/solvothermal crystallization of fine NCs. Those findings were widened by Weller et al. [19,20,21], introducing several competing strategies to control the plasmon peak wavelengths, i.e., by changing the free carrier concentration in ATO NCs. In more detail, the ATO LSPRs were affected by (i) the antimony doping level$\;\frac{N_{\mathrm{Sb}}}{N_{\mathrm{Sn}}}$; (ii) oxidation/reduction of Sb ions, i.e., by balancing the acceptor/donor ($\frac{N_{{\mathrm{Sb}}^{3+}}}{N_{{\mathrm{Sb}}^{5+}}}$) ratio; (iii) by size-control of NCs, i.e., surface/bulk atoms ratio in the NCs indirectly determining the depletion layer thickness; and (iv) additional doping of ATO by oxygen vacancies. The synergy of these four processes allows to tune LSPR pre and during the synthetic procedures, but limits the post-synthetic plasmon control to NC aging effects. Here, we demonstrate the fine-tuning of the LSPR via the choice of ligands in ATO NC solids at the pre- or post-deposition stages, wherein the typical tuning strategy of screening the appropriate dielectric constant of the surrounding medium is generally restricted. We fabricate ATO NCs by the facile solvothermal route; provide their month-scale colloidal stability in polar and nonpolar solvents; and finely tune their LSPRs in spin-casted NC films, enabling the future replacement of their expensive and/or toxic active infrared counterparts.

## 2. Materials and Methods

### 2.1. Chemicals

Metallic granular tin (Sn, 99.5+%), antimony (III) oxide (Sb_2_O_3_, 99%), N-methylformamide (HCONHCH_3_, 99%), antimony(III) chloride (SbCl_3_, ≥99.0%), antimony(III) bromide (SbBr_3_, 99.99%), antimony(III) iodide (SbI_3_, 98%), zinc chloride (ZnCl_2_, ≥98%), iron(II) chloride (FeCl_2_, 98%), methylammonium bromide (CH_6_BrN, ≥99%, anhydrous), lead(II) bromide (PbBr_2_, 99.999%), cobalt(II) chloride (CoCl_2_, ≥98.0%), phosphotungstic acid hydrate (H_3_[P(W_3_O_10_)_4_]·xH_2_O (HPW), reagent grade), phosphomolybdic acid hydrate (H_3_[P(Mo_3_O_10_)_4_]·xH_2_O (HPMO), ACS reagent), indium triiodide (InI_3_, anhydrous, 99.998%), 4-(Hydroxymethyl)benzoic acid (HOCH_2_C_6_H_4_CO_2_H, 99%, BNZ), nitrosyl tetrafluoroborate (NOBF_4_, 95%), and 1,2-ethanedithiol (HSCH_2_CH_2_SH, ≥98.0% (GC)) were received from Sigma-Aldrich (Merck KGaA, Darmstadt, Germany). Oleylamine (OA, CH_3_(CH_2_)_7_CH = CH(CH_2_)_8_NH_2_, 80–90% of C_18_ chains) was bought from Acros Organics (Fisher Scientific GmbH, Schwerte, Germany). Absolute ethanol (ACS grade), nitric acid (ACS grade), n-hexane, tetrachlorethylene (TCE), and toluene were supplied by Merck (Merck KGaA, Darmstadt, Germany, Alfa Aesar, (Thermo Fisher (Kandel) GmbH, Kandel, Germany) and VWR Chemicals, (VWR International GmbH, Darmstadt, Germany) respectively.

### 2.2. Syntheses

The synthesis of Sb_x_SnO_2_ nanocrystals was performed by slightly modifying the procedure proposed by the Gao group [22]. Briefly, metallic tin (0.375 g, 3 mmol) and a required amount of Sb_2_O_3_ (0, 0.075, 0.15, 0.3 mmol; for x = 0 (ATO0); 0.05 (ATO5), 0.1 (ATO10), and 0.2 (ATO20), respectively) were mixed with 9 mL of deionized water, then concentrated nitric acid (5.5 mL) was added dropwise. The mixture was stirred at room temperature until the dissolution of solid precursors appeared (it required c.a. 1–3 h). Afterward, the yellow-colored solution was transferred into a 25 mL Teflon-lined steel autoclave, sealed, and heated at 180 °C for 15 h. The obtained blue (white for the undoped tin oxide) precipitates were separated by centrifugation (10 min at 6000 rpm) and washed twice with deionized water, pure ethanol, and toluene each. At this point, c.a. 200 μL of oleylamine (OA) was added dropwise; a gentle shaking instantly removes the NC solution’s turbidity and makes it optically clear (Appendix A). One can observe that, without the help of amines, the NCs precipitate within minutes. The following NC washing by ethanol, centrifugation, and redispersion was repeated at least three times to remove the excess of OA. Finally, the NCs were redissolved in 5 mL of toluene, hexane, or tetrachlorethylene and filtered through a 0.1 µm PTFE filter.

### 2.3. Ligand Exchange Procedure

The biphasic ligand exchange was a minorly adjusted procedure from [23]. Briefly, 150 µmol of inorganic salts or short organic molecules and 100 mg/mL ATO NC solution were used to obtain concentrated solutions of plasmonic NCs. To the 1 mL solution of inorganic salts in NMF (or DMF for NOBF_4_), 1 mL hexane solution of ATO NCs was added, and the mixture was stirred vigorously for 12 h. Note that the ligand exchange occurs almost instantaneously (for SbCl_3_ within 3 min), but the solutions were left to stir overnight to ensure that the ligand exchange occurs entirely. After that, 1 mL of hexane was added, followed by the subsequent addition of 3 mL of acetone (in the case of NOBF_4_ in toluene), then the reaction mixture was centrifuged for 10 min at 6000 rpm. Washing was performed three times to ensure the complete removal of organic ligands. Finally, Sb_x_SnO_2_ was redispersed in a proper polar solvent (NMF or DMF) with 100 mg/mL concentration. Those solutions are stable for at least one week without observable agglomeration of NCs.

The post-deposition ligand exchange procedure from oleylamine to EDT (1,2-Ethanedithiol) followed by a subsequent oleylamine removal from NC films based on ATO nanocrystals was performed similarly to the procedure reported in [7].

### 2.4. Characterization

XRD patterns were recorded in the 2Θ range 20–70° with a step size of 0.01° from precipitated and vacuum-dried NC powders (c.a. 500 mg) on a Malvern Pananlytical (Kassel, Germany) diffractometer of an Empyrean Series equipped with copper anode operating at 40 kV acceleration voltage and 30 mA current. The obtained data were processed using the JCPDS record #41-1445 for undoped cassiterite tin oxide lattice as a starting iteration.

UV/Vis/NIR absorbance was measured by a LAMBDA 950 spectrometer (Perkin Elmer, Rodgau, Germany) with a PbS IR detector operating from 900 up to 3300 nm. For the measurements, we used diluted (c.a. 5 mg/ mL) NC tetrachlorethylene solutions (in a 10 mm light pass quartz cuvette) and NC films, spin-casted from concentrated (c.a. 100 mg/mL) NC solutions. The procedure was as follows: 125 μm thick glass microscope covers were sequentially washed in an ultrasonic bath by distilled water, acetone, and isopropanol at 60 °C for 10 min. The NC films were triple spin-casted (Ossila spin caster) at 1000 rpm for 30 s. Each casting was followed by 10 min annealing at 110 °C to dense the NCs. For transmission (TEM), scanning electron microscopy (SEM), as well as energy-dispersive X-ray spectroscopy (EDX), we used JEM-1400 Plus and a JEOL JSM7610 instrument (both from JEOL Germany GmbH, Freising, Germany), respectively. Samples for TEM and SEM (EDS) measurements were prepared by drop-casting of diluted NC solution (c.a. 10 mg/mL) on carbon-coated copper grids and silicon wafers, respectively. Differential thermal analysis was performed on a TG 209F1 Libra thermobalance (NETZSCH-Gerätebau GmbH, Selb, Germany). Dynamic light scattering was recorded for NC solutions in PSC115 and ZEN1002 cuvettes using Nano series ZetaSizer from Malvern Pananlytical (Kassel, Germany). X-ray photoelectron spectroscopy (XPS) was performed on drop-casted and dried NC films on Au-coated glass slides with a Theta Probe system from ThermoFisher Scientific, Dreieich, Germany, equipped with a monochromated Al Kα X-ray source (1486,7 eV) and a dual flood gun, which provides low kinetic energy electrons and Ar-ions for efficient charge compensation of the sample surface. The pass energy of the hemispherical analyzer in the constant analyzer energy mode was set to 50 eV for the XPS high-resolution (HR) scans and the energy step size was set to 0.05 eV. An X-ray spot size of 400 µm on the sample surface was chosen for the measurements. Charge referencing of the spectra was obtained by adjusting the contribution of adventitious carbon in the C 1s photoelectron peak to 285 eV. System operation, data evaluation, and peak fitting were performed with the Avantage software package, provided by the manufacturer.

## 3. Results and Discussion

### 3.1. ATO Nanocrystals Exhibiting Localized Surface Plasmon Resonances

The synthesis of ATO nanocrystals starts with the complete dissolution of metallic tin in a mixture of nitric acid and water (Equation (1)), with unreacted antimony (III) oxide at room temperature. No color change or dissolution of the antimony oxide is observed owing to the insufficient reaction temperature required to oxidize Sb^3+^ to Sb^5+^ [24].
4 Sn+10 HNO_3(dilute)_ → 4 Sn(NO_3_)_2_ + NH_4_NO_3_ + 3 H_2_O(1)
2 Sn(NO_3_)_2_) → 2SnO + 4NO_2_ + O_2_(2)
2 SnO + O_2_ → 2SnO_2_(3)
Sb_2_O_3_ + 4 HNO_3_ → Sb_2_O_5_ + 4 NO_2_ + 2 H_2_O(4)

Once the reaction mixture is loaded into an autoclave and the required reaction temperature of 180 °C is reached, tin nitrate (Sn(NO_3_)_2_) is thermally decomposed (Equation (2)) and subsequently oxidized to SnO_2_ in a closed system [25]. At the same time, antimony oxide (Sb_2_O_3_) reacts at room temperature with nitric acid as a hard oxidizer and gives antimony (V) oxide, Sb_2_O_5_ (Equation (4)) [26].

The used solvothermal synthesis of the ATO provides stable colloidal NC solutions with various colors between white and dark blue, dependent on the Sb doping concentration (Figure 1a). The doped (20% antimony) sample’s dimensions were first checked by dynamic light scattering (DLS, Appendix A), providing that the scattering objects have hydrodynamic diameters ranging from 8 to about 40 nm, with the maximum of their size distribution at 11.5 nm. The NC diameter is smaller than that because the aggregation of NCs and the OA ligand shell contribute to their hydrodynamic dimensions. To obtain more precise information, TEM was performed, revealing that the NCs exhibit somewhat irregular shapes (Figure 1b); however, with relatively narrow size distributions (Figure 1c). For instance, the NCs from a batch with a nominal Sb concentration of 20% exhibit a mean size of 5.3 ± 1.3 nm, as manually evaluated from the transmission electron microscopic images. As also found previously [16], the NC size slightly increased for batches with a lower antimony content, and their size distribution became broader (up to 7.1 ± 1.6 nm for undoped SnO_2_ NCs) (Figure 1c).

The elemental compositions of the NCs were evaluated by EDX analysis within several areas of drop-casted ATO (TO) layers prepared on Si substrates (Appendix A). All EDX-spectra provide slightly higher Sb/Sn ratios than used in the syntheses. Over-stoichiometric oxygen can be explained by the storage and deposition of the NCs in ambient conditions; the ligand shell also provides traces of carbon and nitrogen.

According to the international center of diffraction data, the NCs’ crystalline structure was proven to be the cassiterite modification of tin oxide (JCPDS 41-1445). The observed diffraction reflexes (Figure 1d) perfectly match the specific d_hkl_ distances for the orthorhombic SnO_2_ crystal structure (with a = 0.4725 nm and c = 0.3242 nm for the undoped sample). The lattice parameters are only slightly altered for higher tin oxide doping by Sb to a = 0.4717 and c = 0.3231 nm for 20% Sb concentration. Such an almost neglectable dependence of lattice parameters versus the quantity of inserted antimony can be explained, as the host Sn^4+^ ions are substituted simultaneously by both larger (Sb^3+^) and smaller (Sb^5+^) ions [15,27], so that the averaged lattice constant remains almost unchanged. The Sherrer method for the widened XRD peaks (Figure 1d) revealed that crystallite’s mean sizes varied almost linearly from 8.9 nm for undoped SnO_2_ NCs down to 6.3 nm for Sb_0.2_SnO_2_. Those values are close to what we measured from the TEM images.

The localized surface plasmon resonances in extrinsically doped transparent conducting oxide NCs are primarily tunable by the dopant concentration. Thus, we have inspected the series of ATO NC materials synthesized with the different antimony/tin precursors molar ratios. All synthesized NCs (except the undoped tin oxide NCs) are characterized by strong IR absorption, starting at longer wavelengths than that of the absorption edge owing to the bandgap (Appendix A). The NC solution coloration is gradually changed from opal white to several shadows of blue (Figure 1a). Infrared optical absorption reveals the electronic parameters for such heavily doped NCs. The optical absorbance spectra were collected on spin-casted NC films deposited on glass slides and heat-treated to dense NCs (inset in Figure 1e). For the ATO sample with 20% Sb, an explicit resonance peak is observed at a wavelength of 2252 nm (Figure 1e). The peak of the ATO sample with 10% Sb is slightly shifted to the red (to ~2340 nm), in agreement with the observations reported in [28]. Note that increasing the antimony precursor concentration to higher molar rations than 20% (25% and 30%) results in a redshift of the localized surface plasmon resonance, up to 2450 nm (Appendix A). This is attributed to an increased compensation between p- and n-type doping, by an increase in the $\frac{N_{{\mathrm{Sb}}^{3+}}}{N_{{\mathrm{Sb}}^{5+}}}\;$doping ratio, resulting in a decrease in the overall electron concentration. Antimony guest ions substitute tin in the host lattice in different oxidation states by surface Sb^3+^ and Sb^5+^ [16; 27]. As discussed above, higher doping results in smaller NCs, and thus in a higher surface/volume atom ratio, also changing the dopant ratio. The deposition of the NCs on the substrate alters the infrared absorbance properties only slightly, indicating that the dielectric permittivity of the material surrounding the localized surface plasmon hardly changes. This might result from a surface depletion layer surrounding the plasmon confined in the core of the NCs, as described by the Milliron group for highly doped ITO NCs in [29]. This outer depletion layer makes the localized plasmon resonance much less sensitive to the dielectric permittivity of the NCs’ environment. Thus, there is only a small overall redshift of the localized surface plasmon resonance peak after deposition compared with the corresponding resonance in solution (Appendix A). The localized surface plasmon resonance tuning by doping with Sb is limited because of the compensation effect and a surface depletion layer formation. The maximum achieved doping concentration can be determined from the peak position ω_sp_ and full width at half maximum γ of the resonance from the NCs, doped by 20% of Sb, via $N_{car}=\frac{1}{e}\sqrt{\varepsilon_{0}m_{eff}\left(1+2\varepsilon_{m}\right)\left(\omega_{sp}^{2}+\gamma^{2}\right)}$ [30].

Here, ε_m_ is the dielectric permittivity of the plasmon confining medium, which is assumed to be the ATO depletion layer, providing a value of 3.9 [21], and the effective mass of electrons (*m_eff_*) in ATO is calculated in [31]. The estimated free carrier density of 6.7 × 10^20^ cm^−3^ corresponds to 53 electrons per nanocrystals, assuming uniform doping of the SnO_2_. This means that a bit less than 1/5 of the Sb ions incorporated into the tin oxide lattice (about 270 per NC) contributed effectively to free and uncompensated electrons in the conduction band. This ratio decreases compared with NCs, doped by 10% of Sb (51 free electrons and c.a. 135 Sb ions per NC). Thus, with the tuning of free carriers by increasing antimony concentrations (from 10 up to 20%), only one out of the additional (c.a. 68) Sb ions remains uncompensated and contributes to the overall free carrier growth. For the lower doping of 5% of Sb, the localized surface plasmon resonance shifts further into the infrared (Figure 1e) owing to a lower carrier concentration, similar to [32].

### 3.2. Tuning the Localized Surface Plasmon Resonance by Ligand Exchange

Besides increasing the carrier concentration in the bulk of the material by substitutional atoms such as the Sb replacing the Sn, in NCs, the carrier concentrations can also be altered by deposition of charges on the NCs’ surface; for instance, via attachment of appropriate ligands [3,33]. While there are many possible molecules stabilizing ATO in polar solvents [15,16], here, we are restricted to rather “short” species, having the potential to allow electrical transport in NC films. Thanks to their success in quantum dot optoelectronic devices, we have chosen to test several metal-halides (antimony (III) halides SbX_3_, with X = Cl, Br, I; indium (III) iodide InI_3_; transition-metal-dichlorides MCl_2_, with M = Zn, Fe, Co), two acid hydrates (phosphotungstic acid hydrate HPW and phosphomolybdic acid hydrate HPMO), and a (4-(Hydroxymethyl)benzoic acid BNZ). Furthermore, we have tested boron tetrafluoride BF_4_ and 1,2-ethanedithiol EDT, because these ligands have been used to make LSPR NC electrodes in electrochromic windows and field effect transistors [7,34]. The EDT ligands were exchanged after deposition of the NCs on the glass substrates, whereas all others were exchanged in solution. With these small ligands, ATO NCs with an Sb concentration of 20% could be dispersed in polar solvents (Figure 2a), such as DMF and NMF. The short ligands significantly alter the localized surface plasmon resonance peak (Figure 2b). For several ligands, it becomes reduced in intensity and red-shifted in wavelengths (i.e., HPW, HPMO, BNZ, BF_4_) with respect to the resonance observed after heat treating a film of NCs with OA ligands. The observed redshifts with respect to OA capped NCs with the LSPR at 2252 nm (Figure 2) originate from interactions between Lewis-acid ligands and NCs. Such oxidizing ligands withdraw electrons from the NCs, decreasing the NCs’ free carrier concentrations and lowering their colloidal stability [35].

Some ligands change the color of the colloidal solution more drastically than HPW, HPMO, BNZ, or BF_4_, from bluish (ZnCl_2_) into dark green-black (FeCl_2_) or blue-black (CoCl_2_) (Figure 3a). After film deposition, however, their LSPRs exhibit roughly the same amounts of red shifts as those exhibited by the other Lewis acid ligands (Figure 3b), suggesting that electrons have been withdrawn from NCs as effectively as by the conventional oxidizing ligand BF_4_ [36]. The stronger coloration of the colloidal solutions might be caused by some residuals of the metal-chloride salts used for ligand exchange, being colored themselves.

In contrast to the ligands discussed above, the ethanedithiol ligands represent a weak Lewis base, and the treated NC film exhibits a minimal blue shift of the localized surface plasmon resonance (Figure 2b) regarding the OA capped NCs. This effect is even more pronounced for antimony-halides (SbX_3_, X = I, Br, Cl), resulting in a larger blueshift of the LSPR of up to 100 nm and an enhancement of the plasmon intensity (Figure 2b and Appendix A). Similar blueshifts of LSPRs have also been obtained in ATO NCs, initially doped by 10% antimony. These blue shifts are enabled because, in trivalent antimony halides, two outer *s* and one *p* electron of Sb are bonded by three halide anions so that two unpaired *p*-electrons remain, which can be provided to the NCs, where they might also increase the free carrier concentration. Using the chemical geometry optimization builder Avogadro [37], the possible NC–ligand interactions resulting in electron donation are sketched in Figure 4a.

The Sb^3+^ ions of the antimony trichloride ligands donate their lone *p*-electron pair when attaching to the NCs’ surface (Figure 4a). When attached at the surface, a charge transfer transition between Sb^3+^ and Sb^5+^ may occur [18], accommodating additional electrons within the NC. The other ligand providing a blue-shifting of the localized plasmon resonance, namely EDT, behaves similarly to long-chain thiols reported in [38]. The thiol molecule donates the sulfur lone electron pair to the lattice cation, thus increasing the NCs’ electron density. A sketch of the opposite situation, where the ligand exchange results in a redshift of the localized surface plasmon resonance by decreasing the electron density in the NCs, is shown in Figure 4b. The red-shifting ligands are conventional Bronsted–Lowry or Lewis acid-type of molecules (simply called “acids” in Figure 2). Acidic ligands coordinated by polarized hydroxyl groups bind to the electron enriched and undercoordinated Sb^3+^ ions on the NCs’ surface, similarly as described in [38]. The depleted hydrogens are attracted and trap the lone electrons of the Sb^3+^, initiating an Sb^5+^→Sb^3+^ transition (Figure 4b) to rebalance the NCs’ charge neutrality, thus decreasing the overall NCs’ electron concentration. In the proton-free Lewis acid ligands, e.g., in InI_3_, the electron-depleted indiums are in their highest possible oxidized form (likewise for the other metals in halides, Figure 3 and Appendix A), thus also accepting electrons from an NC. This is similar to the case of LSPRs in ITO NCs when the surface is functionalized by the tungsten (VI) complexes [34], also resulting in a redshift of plasmon transitions. When comparing the donation abilities of different antimony halides (Appendix A), the smaller chemical potential of reducing ligands causes less electron density to be transferred to the NCs [35]. Considering that chemical potential is the negative value of the molar Gibbes free energy reported in [39] for SbCl_3_: Δ*_f_G*^0^ = −323 kJ/mol, SbBr_3_: Δ*_f_G*^0^ = −259 kJ/mol, and SbI_3_: Δ*_f_G*^0^ = *c.a.*−100 kJ/mol, explains the higher reducing ability of antimony chloride than bromide and iodide. In other words, the chlorine ions electrostatically repulse the lone electron pair of antimony more than bromine and iodine. In fact, for the SbCl_3_ ligands, we found the highest blue shift of the LSPR among all tested ligands, which is thus an extreme case. The ligand exchange procedure to SbCl_3_ neither changed the NCs’ dimensions (Figure 5a–d), nor caused a shift in the diffraction reflexes, nor widened them (Figure 5e). Thus, surface antimony from ligands provides additional charges without incorporating into the NCs’ bulk, and does not forms an epitaxial shell of thicknesses to be detected by XRD. As stated already above, the charge transfer from the ligands to the NCs operates not only for the 20% doped samples, but also for those with a lower Sb concentration (Figure 5f).

To judge the efficiency of the charge transfer doping from the ligands towards the NCs, the number of attached ligands have to be quantified. For that purpose, TGA measurements were performed and analyzed (Appendix A). In detail, the grafting density, i.e., the number of molecules on the particle surface, was calculated as follows:(5)$$GD=\left(\frac{wt}{100-wt}\right)\left(\frac{N_{A}}{MW*SSA}\right)$$
where *wt* stands for the percentage of mass loss, N_A_ (6.022 × 10^23^ mol^−1^) for the Avogadro constant, *MW* for the molecular weight of the surfactant, and *SSA* for the specific surface area in nm^2^/g of nanoparticles. The *SSA* was determined under the assumption of spherical particles with an average diameter of 5.3 ± 1.3 nm (measured by TEM) and a density of 6.8 g/cm^3^. The numbers in Table 1 suggest that the quantity of SbCl_3_ ligands is ~80% of that of the initial OA ligands.

By evaluating the carrier concentrations from the measured plasmon peak energies, the observed blue shifts result clearly from doping via charge transfer from the ligands to the NCs. The comparison of the number of SbCl_3_ ligands attached with the increase in carriers per nanocrystal evidences that a relatively high portion of them contributed to the electron donation to the ATO NC free carriers. Indeed, the overall number of carriers per NC increased by up to six excess electrons during ligand exchange from OA to SbCl_3_. This well correlates with the data from [36], wherein the LSPR NCs of comparable mean sizes underwent ligand exchanges from long OA to short inorganic ligands of an oxidizing/reducing nature. Thus, the doping efficiency of surface antimony chloride is about 3.5%. The rest of the ligands are probably required to maintain the colloidal stability of the NC solutions, which would not be the case if a substantially higher number of ligands would take part in the electron transfer to the NCs [33]. The doping efficiency of 3.5% from the antimony ligands is considerably lower than the averaged doping efficiency obtained by aliovalent doping. However, as discussed above, increasing the Sb concentration by aliovalent doping to >20% results in an effective decrease in the electron concentration owing to compensation effects, thus the doping efficiency becomes a negative value. Doping by ligands representing Lewis bases is apparently not restricted as much as by dopant compensation, and allows an increase in electron concentration further than that by aliovalent doping.

Finally, we want to mention that doping of tin oxide by Sb ions is strongly correlated to the difference between Sb^5+^ and Sb^3+^. Within bulk, Sb^3+^ represents an acceptor occupied by an electron from the valence band and Sb^5+^ represents an ionized donor. Thus, quantifying Sb^3+^ to Sb^5+^ ratios or better differences should correlate to the observed carrier concentration. Attaching SbCl_3_ ligands complicates this evaluation because, in this case, Sb^3+^ is present at the surface if no charge transfer occurred to the NCs. By XPS, different spectra are thus obtained for the NCs before and after ligand exchange (Appendix A), which basically confirms the results from the EDS (Appendix A) and evidences the successful ligand exchange by quantifying the increase in the Sb concentration after ligand exchange. Unfortunately, however, in the XPS spectra, the spin-orbit split part of the core Sb level, namely the peak resulting from Sb 3d_5/2_, overlaps with the O 1s peak, hampering any quantification of oxygen atoms bound to Sb^3+^ or Sb^5+^ [16,40]. There is also a less intense peak originating from the Sb 3d_3/2_ spin state (Appendix A), which exhibits an asymmetric shape for the ATO NCs with a concentration of 20%, similar to that in [15,16]. The lower energy band is frequently correlated to Sb^3+^ and the one at a higher energy to Sb^5+^, irrespective of their accurate peak energies. However, in a recent publication of C. Liu et al. [41], an alternative assignment is presented. From a comparison of the principal peak with the known binding energies measured in Sb_2_O_5_, Sb_2_O_4_, and Sb_2_O_3_, it is concluded that this peak has to be ascribed to a mixed Sb^3+^/Sb^5+^ valency and the side peak at a higher energy is due to a disordered many-body-effect screening mechanism [42,43]. Such a many body-effect can also be the free electrons in the conduction band, causing the plasmon resonance, which is directly observed here in the infrared transmittance.

## 4. Conclusions

The solvothermal synthesis of ATO provides nanocrystals with nm sizes, exhibiting LSPRs in the infrared, dependent on the doping level. By post-synthetic ligand exchange, not only colloidal stability is achieved, but also tuning of the LSPR energy. Dependent on the chosen type here, the ligand exchange is demonstrated to provide electron transfer, to and from the nanocrystals, resulting in red and blue shifts of the LSPR. The ligand exchange not only allows the fine-tuning of the LSPR energy, but also to obtain doping levels that are not accessible by aliovalent doping of the nanocrystal inorganic core materials. In that respect, the ligand exchange is a powerful tool for LSPR shifting and carrier concentration tuning. The tuning and doping presented here for ATO is certainly not restricted to this particular metal oxide, but will also be adaptable for other nanocrystal species, especially plasmonic ones. Plasmonic tuning, as demonstrated here, could find numerous applications in infrared optical devices, as developed in innovative glazing technologies, for harvesting wasted parts of the solar’s infrared spectrum in energy conversion devices, or applied in infrared driven therapeutics, at least when performed in the medical infrared windows of the light spectrum for which tissues are transparent.

## Figures and Tables

**Figure 1 nanomaterials-12-03469-f001:**
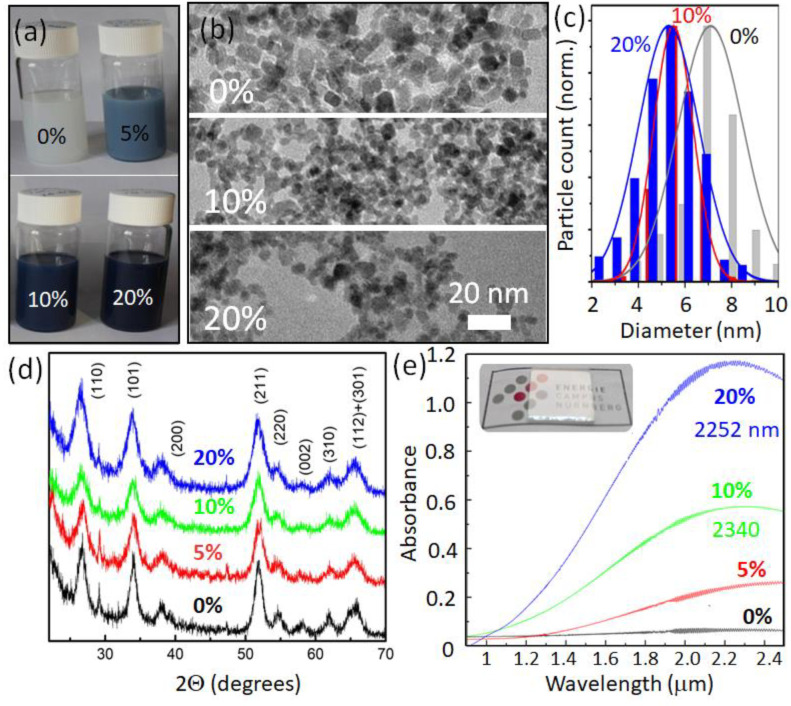
Synthesized ATO nanocrystals. (**a**) Photo of the colloidal solutions of OA capped NCs for various doping concentrations given in atomic per cents (100 mg/mL in toluene). (**b**) TEM images, (**c**) size distribution histograms, (**d**) XRD patterns of ATO NCs with various nominal Sb concentrations, and (**e**) infrared absorbance of ATO NC films (a typical film on a glass substrate is shown in the inset). Next to the nominal doping concentrations, the LSPR resonance wavelengths are provided for the 20% and 10% curves.

**Figure 2 nanomaterials-12-03469-f002:**
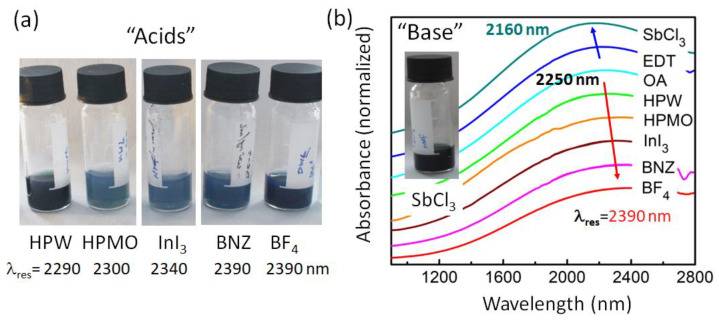
ATO NCs with an Sb doping concentration of 20% after ligand exchange to electron withdrawing (“acids”) and electron donating (“bases”) small ligands. (**a**) Photo of the colloidal solutions after exchanging OA ligands by the ligands indicated below the photos. Given are also the measured resonance wavelengths of the LSPRs. (**b**) Infrared absorbance spectra of NCs’ films with various ligand species on a normalized scale. For clarity, the origins of each spectra are shifted in the y-direction. The inset shows a photo of NCs covered by SbCl_3_ as a ligand.

**Figure 3 nanomaterials-12-03469-f003:**
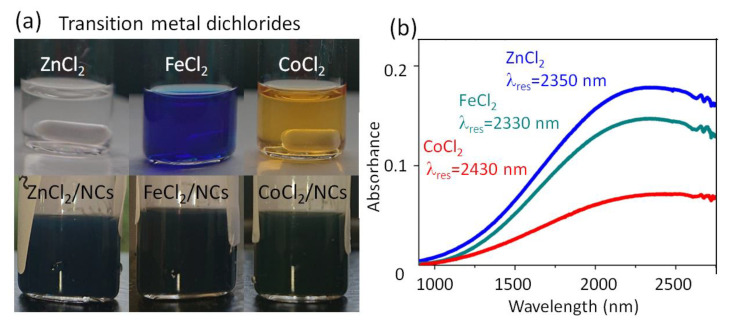
ATO NCs with a Sb doping concentration of 20% after ligand exchange to transition metal dichloride ligands. (**a**) Photo of ZnCl_2_, FeCl_2_, and CoCl_2_ dissolved in NMF (upper row) and of the NCs after completed ligand exchange (lower row). (**b**) Infrared absorbance spectra of NCs’ films with various ligand species, also providing the LSPR wavelengths.

**Figure 4 nanomaterials-12-03469-f004:**
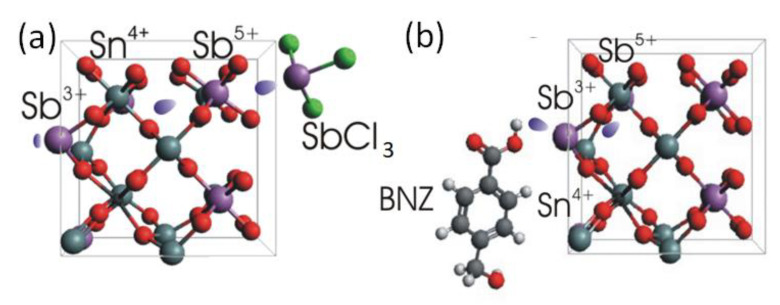
Sketches of the charge transfer via typical donating and accepting ligands attached to the ATO NC surfaces: optimized geometry of (**a**) SbCl_3_ and (**b**) BNZ ligands in the stick-ball model and tentative electron density transfers.

**Figure 5 nanomaterials-12-03469-f005:**
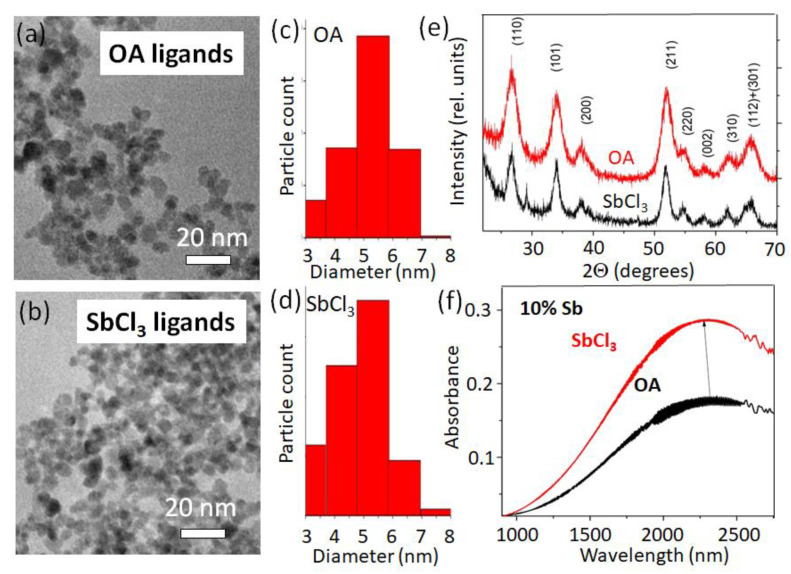
ATO NCs upon ligand exchange to SbCl_3_. (**a**,**b**) TEM images of 20% doped ATO NCs before and after ligand exchange with (**c**,**d**) their corresponding size distributions and (**e**) X-ray diffraction spectra. (**f**) LSPR shift of ATO NCs with a 10% doping concentration upon ligand exchange.

**Table 1 nanomaterials-12-03469-t001:** Ligand attachment and its effect on carrier concentration.

Ligand	Mass Loss (%)	Ligand Density (1/nm^2^)	Ligand Number (1/ NC)	Electrons (cm^−3^)	Electrons (1/NC)
SbCl_3_	10.95	1.95	172	7.6 × 10^20^	59
OA	15.79	2.54	224	6.8 × 10^20^	53

## Data Availability

After acceptance of the manuscript, the data will be made available for open access via the OPUS FAU depository of our university.

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
