# Peer review of "Ligand Tuning of Localized Surface Plasmon Resonances in Antimony-Doped Tin Oxide Nanocrystals"

_nanomaterials, 2022, doi:10.3390/nano12193469_

Round 1
Reviewer 1 Report
This manuscript by Balitskii and coauthors demonstrated a synthetic approach of antimony doped Tin oxide nanocrystals which exhibit surface plasmon resonances depending on the doping level. In addition, post-synthetic ligand exchange was employed to tune the resonance peak, where they found that electron withdrawing ligands can cause red shift while electron donating ligands could cause blue shift on the LSPR. This manuscript is well written and presented with deep discussions. The conclusions are supported the by the experiments and results. I would like to suggest Nanomaterials accept this manuscript. I have only few comments as below for the authors.
1. Is this a typo in line 193 “The undoped (20% Antimony) samples…”? Should be 20% doped sample?
2. Please make sure the plot color consistent in Figure 1d and 1e; for example, both using blue for 10% and green for 20%?
3. Do authors has a EDX mapping of the nanocrystals which can show the homogeneous distribution of the elements over the NC volume.
4. Do authors have TEM images with better quality for 10% and 20% doped ATO? I can see shaped particles from 0% sample, but 10% and 20% images more look like amorphous due to the low resolution.
5. The downward arrow in Figure 2b is confusing, which seems indicate a continuous red shift of the λres. Though, I only see a continuous intensity decrease with those acidic ligands. The resonance peak maximum seems not to be proportional to the intensity. I was assuming that red shift of the peak maximum should accompany with decreasing in peak intensity due to the decrease in carrier concentration. Though, BF4 gives most red shift but still highest intensity among those Lewis acid ligands. Is there any explanation of the relationship between LSPR peak position and intensity?
Author Response
Answers to the comments of the reviewers 1
Reviewer 1, general statements: This manuscript by Balitskii and coauthors demonstrated a synthetic approach of antimony doped Tin oxide nanocrystals which exhibit surface plasmon resonances depending on the doping level. In addition, post-synthetic ligand exchange was employed to tune the resonance peak, where they found that electron withdrawing ligands can cause red shift while electron donating ligands could cause blue shift on the LSPR. This manuscript is well written and presented with deep discussions. The conclusions are supported the by the experiments and results. I would like to suggest Nanomaterials accept this manuscript. I have only few comments as below for the authors.
Reply to reviewer 1, general statement: We thank reviewer 1 for his contributions to the papers and his helpful comments. We have done corrections according to his advices.
Reviewer 1, comment 1: Is this a typo in line 193 “The undoped (20% Antimony) samples…”? Should be 20% doped sample?
Reply to reviewer 1, comment 1: Thank you very much for detecting this type. We have corrected it.
Reviewer 1, comment 2: Please make sure the plot color consistent in Figure 1d and 1e; for example, both using blue for 10% and green for 20%?
Reply to reviewer 1, comment 2: We have changed Figure 1 d) to fit the sequence of colors to those used in Figure 1 e). Please check the figure in the manuscript with marked changes.
Reviewer 1, comment 3: Do authors has a EDX mapping of the nanocrystals which can show the homogeneous distribution of the elements over the NC volume.
Answer: to reviewer 1, comment 3: We are very sorry, but we cannot provide EDX maps across individual nanocrystals. We wonder also how representative this could be, considering that all the nanocrystals are not homogeneous ins size and shape.
Reviewer 1, comment 4: Do authors have TEM images with better quality for 10% and 20% doped ATO? I can see shaped particles from 0% sample, but 10% and 20% images more look like amorphous due to the low resolution.
Answer to reviewer 1, comment 4: We have replaced the TEM images by ones done with higher magnification. We think that they show more details from the nanocrystals, but are also not perfect. Possibly the nanocrystals are covered with some excess of organics (ligands) which make obtaining better TEM images difficult. We also did not get access to a high-end TEM machine for this project. The new figure please check in the manuscript with marked changes.
Reviewer 1, comment 5: The downward arrow in Figure 2b is confusing, which seems indicate a continuous red shift of the λres. Though, I only see a continuous intensity decrease with those acidic ligands. The resonance peak maximum seems not to be proportional to the intensity. I was assuming that red shift of the peak maximum should accompany with decreasing in peak intensity due to the decrease in carrier concentration. Though, BF4 gives most red shift but still highest intensity among those Lewis acid ligands. Is there any explanation of the relationship between LSPR peak position and intensity?
Answer to reviewer 1, comment 5: We completely agree to this statement and thus have changed the figure. For that purpose, we have done a new smoothing of the data and brought them in order of the blue and red shifts. The statement of the reviewer is correct, that the decrease of intensity should also correlate to the observed red shift, because both quantities are related to the electron concentration. Different intensities, however, can be observed when the obtained film thickness is not the same for all samples or if the film thickness is inhomogeneous. To avoid here confusions, we have plotted know normalized data. These data are shifted in y-direction for better clarity, which is also noted in the figure captions. The new figure is shown in the manuscript. In addition, we found that one of the resonance wavelengths given the previous version was a misprint and we corrected this value.
Reviewer 2 Report
This article has aroused my interest. In this paper, a comprehensive study on plasmonic ATO nanocrystals: preparation, electronic and LSPRs properties were successfully studied. It was interesting that the authors demonstrate aliovalent doping and ligand exchange can tune the LSPR energies of ATO. The purpose and the quality of the experiment are good. And the results are of interest to other researchers. Therefore, I am glad to see this manuscript published in Nanomaterials after minor revisions.
1. Could you explain the peak shift in Figure 1d? More detailed information is highly required.
2. Could you check whether some vacancies appear after doping?
3. Could you provide an HRTEM image for a clear lattice?
4. The manuscript should be carefully checked, There are some obvious mistakes, such as “ LPRs” (Line 31 page 1), and “180oC” (Line 187 page 4). The reference format used in the text is incorrect. Please modify it. For example, “zum Felde, U.;” should be “Zum Felde, U.;”. The concentrations of transition metal dichloride ligands need to be added. From line 329 to Line 258, the format size was an error.
5. Recently, there have been many articles related to this field, such as Chemosphere 302 (2022): 134849; Energy Storage Materials 32 (2020): 167 -177; Chemical Engineering Journal 430 (2022): 132829; which need to be considered.
Author Response
Answers to the statements of reviewer 2
Reviewer 2, general statement: This article has aroused my interest. In this paper, a comprehensive study on plasmonic ATO nanocrystals: preparation, electronic and LSPRs properties were successfully studied. It was interesting that the authors demonstrate aliovalent doping and ligand exchange can tune the LSPR energies of ATO. The purpose and the quality of the experiment are good. And the results are of interest to other researchers. Therefore, I am glad to see this manuscript published in Nanomaterials after minor revisions.
Answer to reviewer 2, general statement: Thank you very much for reviewing our work and for your valuable contributions. We attempted to improve the manuscript according to your suggestions.
Reviewer 2, comment 1: Could you explain the peak shift in Figure 1d? More detailed information is highly required.
Answer to reviewer 2, comment 1. Yes, we can give more details. We write now: “The observed diffraction reflexes (Fig 1d) match perfectly the specific dhkl distances for the orthorhombic SnO2 crystal structure (with a=0.4725 nm and c=0.3242 nm for the undoped sample). The lattice parameters alter only slightly for higher tin oxide doping by Sb to a=0.4717 and c=0.3231 nm for 20% Sb concentration. Such an almost neglectable dependence of lattice parameters vs the quantity of inserted antimony can be explained, as the host Sn4+ ions are substituted simultaneously by both, larger (Sb3+) and smaller (Sb5+) ions [15, 27], so that the averaged lattice constant remains almost unchanged.
Reviewer 2, comment 2: Could you check whether some vacancies appear after doping?
Reply to reviewer 2, comment 2: Honestly, we do not know any method which allows to check the appearance of vacancies in the presence of Sb in ATO. We attempted to synthesize SnO2 with vacancies, but did not find any signature of plasmons in that material.
Reviewer 2, comment 3: Could you provide an HRTEM image for a clear lattice?
Reply to reviewer 2, comment 3: Unfortunately, we did not get access to a TEM providing high resolution images of our material. We, however, also do not see the relevance for such an investigation since we doubt how representative that would be in view of the relatively large size distribution and shape distribution of the synthesized nanocrystals.
Reviewer 2, comment 4: The manuscript should be carefully checked, There are some obvious mistakes, such as “ LPRs” (Line 31 page 1), and “180oC” (Line 187 page 4). The reference format used in the text is incorrect. Please modify it. For example, “zum Felde, U.;” should be “Zum Felde, U.;”. The concentrations of transition metal dichloride ligands need to be added. From line 329 to Line 258, the format size was an error.
Answer to reviewer 2, comment 4: Thank you very much for highlighting the mistakes which we have corrected. We found also some additional ones which we have corrected. Just to the writing of “zum Felde” we want to say, that this writing is possibly correct. Here “zum” is not a name but an indicator that the name is an aristocratic one. I think this is used only in German and in German this “zum” has to be written with a small “z”. The concentration of ligands for the transition metal dichloride ligands is the same as for the others. This value is provided in the experimental section (line 133).
Reviewer 2, comment 5: Recently, there have been many articles related to this field, such as Chemosphere 302 (2022): 134849; Energy Storage Materials 32 (2020): 167 -177; Chemical Engineering Journal 430 (2022): 132829; which need to be considered.
Answer to reviewer 2, comment 5: Thank you for highlighting these papers which are all interesting. (Chemosphere (Impact Factor 7.086), Volume 302, September 2022, 134849, Activation of persulfate via Mn doped Mg/Al layered double hydroxide for effective degradation of organics: Insights from chemical and structural variability of catalyst, Energy Storage Materials (Impact Factor 18.80), Volume 32, November 2020, Pages 167-177, Structural engineering and surface modification of MOF-derived cobalt-based hybrid nanosheets for flexible solid-state supercapacitors, Chemical Engineering Journal (impact factor 13.273), Volume 430, Part 2, 15 February 2022, 132829, On-site H2O2 electro-generation process combined with ultraviolet: A promising approach for odorous compounds purification in drinking water system). We have checked them but find them not that much related to the present manuscript so that we do not cite them.